# Impact of Pesticide Residues on the Gut-Microbiota–Blood–Brain Barrier Axis: A Narrative Review

**DOI:** 10.3390/ijms24076147

**Published:** 2023-03-24

**Authors:** Maria Abou Diwan, Marwa Lahimer, Véronique Bach, Fabien Gosselet, Hafida Khorsi-Cauet, Pietra Candela

**Affiliations:** 1PERITOX—Périnatalité et Risques Toxiques—UMR_I 01, Centre Universitaire de Recherche en Santé, CURS-UPJV, University of Picardy Jules Verne, CEDEX 1, 80054 Amiens, France; maria.abou.diwan@u-picardie.fr (M.A.D.); marwa.lahimer@etud.u-picardie.fr (M.L.); veronique.bach@u-picardie.fr (V.B.); hafida.khorsi@u-picardie.fr (H.K.-C.); 2Laboratoire de la Barrière Hémato-Encéphalique (LBHE), UR 2465, University of Artois, 62300 Lens, France; fabien.gosselet@univ-artois.fr

**Keywords:** gut-microbiota-BBB axis, barriers, pesticides, Chlorpyrifos, prebiotics, vulnerable population

## Abstract

Accumulating evidence indicates that chronic exposure to a low level of pesticides found in diet affects the human gut-microbiota–blood–brain barrier (BBB) axis. This axis describes the physiological and bidirectional connection between the microbiota, the intestinal barrier (IB), and the BBB. Preclinical observations reported a gut microbial alteration induced by pesticides, also known as dysbiosis, a condition associated not only with gastrointestinal disorders but also with diseases affecting other distal organs, such as the BBB. However, the interplay between pesticides, microbiota, the IB, and the BBB is still not fully explored. In this review, we first consider the similarities/differences between these two physiological barriers and the different pathways that link the gut microbiota and the BBB to better understand the dialogue between bacteria and the brain. We then discuss the effects of chronic oral pesticide exposure on the gut-microbiota-BBB axis and raise awareness of the danger of chronic exposure, especially during the perinatal period (pregnant women and offspring).

## 1. Introduction

Given the sharp rise in the incidence of cancer, leukemia, inflammatory bowel disease (IBD), and neurological diseases, it is evident that human chronic exposure to pesticides is a real public health problem [1,2]. Since 1975, the World Health Organization (WHO) has classified families of pesticides as “hazardous substances” and has continued to classify them according to their dangerousness in the light of the latest knowledge [3,4]. Despite the effort to limit their use, pesticides are widely used throughout the developing world, and their demand is increasing due to the current system of crop production which prioritizes high agriculture yields [5]. Pesticides have generally been considered to be contributors to global food security and were originally designed to kill pests by interacting with the targets involved in vital functions (e.g., nerve signaling, metabolism, cell survival, or division). However, as the research in recent years has shown, they can have deleterious effects on human health (nontargeted organism) by modifying physiological mechanisms that are not specific to them [6,7]. To limit the entry of potential hazards and to protect itself from pathogens, environmental toxins, or neurotoxic molecules, the body has developed several barriers, including epithelial cell barriers (the intestinal barrier (IB)), the air–blood lung barrier, the blood–cerebrospinal fluid barrier (BCSF-B), and the blood–brain barrier (BBB), each displaying different barrier properties and cellular composition [8,9,10]. The IB and BBB play an important role by protecting the host against food, environmental contaminants, and infections. The IB, located in the intestine’s surface, is part of the first defense system between the external environment and the internal systems of our body [11]. This barrier is built from a single layer of epithelial and Paneth cells separating the gut lumen from the internal space, whereas the BBB is a selective barrier localized at the endothelial cells (ECs) constituting the brain microvessels, which separates the lumen of blood vessels from the central nervous system (CNS) parenchyma [12,13,14]. The IB and BBB are considered immunological and physical barriers [15,16,17]. Each barrier not only provides protection against invading pathogens but is also important for controlling the microenvironment of the tissue and, therefore, tightly regulates the movement of the molecules and ions between the cellular spaces [13,18,19]. These barriers have many similarities in their mechanisms of action despite providing defense in very different environments [20]. Unlike the BBB, the IB is constantly exposed to food antigens and contaminants and is colonized by a collection of bacteria and microorganisms’ antigens of the microbiota. The gut microbiota (GM) is a real organ system that includes a diverse and complex population of microorganisms colonizing the digestive tract and having a symbiotic host’s relationship that helps to maintain a dynamic metabolic and ecological balance [21]. In fact, the GM consists of more than 3 million genes compared to 23,000 genes in the human genome, making the microbiome crucial in many functions in the human organism’s health and disease [22]. The neural, endocrine, and immune network of communication exists between the gut, microbiota, and brain. This axis of bidirectional communication, defined as the gut-microbiota–brain axis is essential in maintaining homeostasis of the gastrointestinal, CNS, and microbial systems (Figure 1) [23,24].

Under normal, healthy conditions, mucus and a tight barrier of epithelial cells confine most microbes to the gut lumen [12]. Daily exposure to contaminants, including pesticides, changes the structure and the balance of the GM (named gut dysbiosis or dysbacteriosis) in favor of potentially pathogenic bacteria [25,26]. Dysbiosis is defined as an imbalance in the GM and/or its functions due to the loss of beneficial microbes and the increase in pathogenic ones that might induce inflammation and immune dysregulation leading to an unhealthy outcome [27,28,29]. As a result, the alteration in bacterial metabolites modulates the IB permeability and thus generates an uncontrolled and permissive passage of not only proinflammatory agents (cytokines, interleukins, bacteria, and bacterial products) [30,31] but also bacterial neurotransmitters from the microbial environment and immune cells that can compromise intestinal and brain homeostasis by affecting the BBB integrity leading to inflammatory and neurological disorders [32,33]. The passage of viable bacteria of the gastrointestinal flora is through the barrier of the intestinal mucosa (the lamina propria) to the mesenteric nodes and then to normally sterile internal organs and is defined as “bacterial translocation” [26].

The link between pesticides and diseases is not limited to the dose we are exposed to but most importantly (1) the timing of the exposure (prenatal exposure, the first 1000 days of life) or “windows of vulnerability” of the Developmental Origins of Health and Disease (DOHaD) concept and (2) the duration of the exposure: a chronic exposure to a low dose or to the dose with no effect (No-observed-adverse-effect level: NOAEL) can have more of an impact on health than a short-term exposure to a high dose [26,33,34,35,36]. Therefore, understanding how the gut, microbiota, and BBB are affected by environmental factors such as pesticides is important for elucidating the way of preventing and treating bowel and brain pathologies.

Nowadays, the interplay between pesticides, the GM, and the BBB is still not fully explored. Consequently, to understand how the IB and BBB are affected following microbiota dysbiosis induced by pesticides, it is necessary to (a) acknowledge the structure and the function of these three “organs” as if they form one system which is the gut-microbiota-BBB axis and analyze how pesticides could impact this axis.

We will first discuss the similarities and the differences between the IB and the BBB. Then, we will focus on the current knowledge of the effects of pesticides on this axis and raise awareness of the danger of chronic exposure, especially during the perinatal period. Finally, we will briefly discuss whether prebiotics could counteract the effects of these xenobiotics.

## 2. Formation, Composition, and Role of Gut Microbiota

Most of the research on the GM assesses that microbial colonization of the human gut begins at birth but progressively evolves and is modified by surrounding factors, such as environment and diet [37]. The process of colonization of the gastrointestinal tract (GIT) is influenced by the type of birth, which means that microbial species of C-section neonates differ from those of vaginally born infants [38]. Advanced studies showed the presence of microorganisms in the meconium of individuals born by cesarean section and in the umbilical cord blood of newly born babies, which means that there is microbiota transmission from the mother to the fetus and thus the fetus does not live in a fully germ-free mother womb [21,39,40,41]. The GM is extremely complex, and its composition changes after the shift from breast-feeding to solid food and is continuously influenced by numerous host-related factors that are external (environment and diet: type of food and feeding habits) and internal (intestinal pH, microbial interactions, temperature, peristalsis, bile acids, intestinal secretions, and immune responses). At adulthood, the GIT contains a vast and complex microbial ecosystem of approximately 100 trillion microorganisms with more than 40,000 species of bacteria containing 100 times more genes than humans [42,43]. Most of these bacteria protect the gut epithelial cells against pathogens. Strict anaerobes mainly compose the GM and outnumber the facultative anaerobes and the aerobes by up to 100-fold [44,45]. Out of the four phyla populating the gut, two phyla, *Firmicutes* and *Bacteroidetes*, appear predominant in the human GM [46,47]. Despite all the factors that influence the composition of the intestinal microbiota, the microbial community stays stable at the phylum level: the two phyla mentioned above are conserved in individuals and may only vary in their relative proportions at various stages in life [47,48,49]. However, on the spatial level of the GIT, the phyla distribution varies. Biopsy studies showed that the small intestine is enriched with certain members of the *Firmicutes* phyla and the colon with members of the phylum *Bacteroidetes* [50,51]. Variations also exist between intestinal sections. For example, the genera *Bacteroides*, *Bifidobacterium*, *Streptococcus*, *Enterococcus*, *Clostridium*, *Lactobacillus*, and *Ruminococcus* were all found in the feces, making the composition representative of the luminal community, whereas only *Clostridium*, *Lactobacillus*, and *Enterococcus* were detected in the mucus layer and epithelial crypts of the small intestine [46].

The GM is a major component of the digestive tract that stimulates the regeneration of intestinal epithelial cells and mucus production by goblet cells. It similarly has a role in nutrient assimilation and the fermentation of non-digestible substrates [29].

Host immunity maintenance depends on the gut microbial–host interactions because 70% of the immune system resides in the gut. The GM stimulates the innate immune system early in life leading to the maturity of gut-related lymphoid tissue and inspires acquired immunity by stimulating local and systemic immune responses, gut synthesis, and the metabolism of certain nutrients [52,53]. The GM nourishes the intestinal mucosa by producing short-chain fatty acids (SCFAs): acetic, propionic, and butyric acids [54]. Acetate and propionate are produced by *Bacteroides thetaiotaomicron* (*Bacteroidetes* phylum), whereas butyrate is produced by *Clostridium tyrobutyricum* (*Firmicutes* phylum). A study showed that SCFAs are able to increase the number and function of regulatory T cells (Tregs) in the gut [55]. SCFAs are also able to decrease BBB leakage: a study on monocolonized mice with single bacteria strain *Clostridium tyrobutyricum* or *Bacteroides thetaiotaomicron* showed a decrease in BBB permeability compared to germ-free mice. Moreover, when germ-free mice, having disorganized brain tight junctions (TJs), were colonized by the microbiota of pathogen-free mice, the integrity of the BBB was increased, proved by the increase in TJs expression: occludin and claudin-5 [56].

In addition to SCFAs, the GM produces many metabolites, such as lipopolysaccharides (LPS), trimethylamine (TMA), bile acids, and indoles. [57]. Some of these metabolites are beneficial such as indoles that regulate the IB function and immune response, and bile acids and TMA that maintain glycolipid homeostasis. Contrarywise, others are detrimental such as LPS which is an immune system disrupter and circulating uremic toxins derived from the digestion of dietary amino acids that have recently been shown to impair BBB integrity and alter brain activity (p-cresol sulfate and 4-ethylphenyl sulfate) [58,59].

## 3. The Structure and Functions of the Intestinal Barrier

As mentioned above, the IB refers to the barrier between the microbiota and the intestinal mucosa and is one of the largest boundary barriers between the body and its environment [60,61].

The IB constitutes a biochemical and physical barrier where millions of microbes and environmental antigens come in close contact with the host immune system. The IB provides an effective obstacle and becomes more selective at birth [62]. This barrier is made of a mucus layer that represents the first defense component of the IB, limiting the transport of potentially harmful antigens and microorganisms. The intestinal mucosa is composed mainly of mucus, highly glycosylated mucin proteins, defensins, Immunoglobulin A (IgA), and the inner lamina propria where regulatory T cells (Tregs) and resident dendritic cells maintain an anti-inflammatory environment by secreting an appropriate cocktail of cytokines [63,64]. This layer, essential in the maintenance of intestinal homeostasis, coats the interior surface of the GIT and acts as a physical barrier to bacteria and other antigenic substances present from the lumen to the lamina propria [63].

Despite its structural role, the mucus layer does not establish alone a significant barrier to transmucosal water or solute flux. The IB depends on the epithelial cells that are the bricks of the intestinal physical barrier. The latter is associated with absorptive enterocytes, goblet cells, enteroendocrine cells, Paneth cells, and microfold cells [65]. Some of these cells release neuro-immunomodulatory mediators (e.g., enterochromaffin cells) such as serotonin, while others such as Paneth cells release anti-microbial peptides [66]. These cells all together form a continuous and polarized monolayer separating the gut lumen from the internal space.

The passage of most hydrophilic solutes through the adjacent epithelial cells is highly restricted. Indeed, the paracellular route of solutes, ions, and nutrients is regulated through the presence of junctional complexes providing a physical barrier to unwanted and potentially damaging molecules [18]. The function of the TJs is important in intestinal health as a defective IB leads to diseases, including bacterial enteritis, IBD, and irritable bowel syndrome (IBS) [67].

From apical to basal, the intercellular space is sealed by a complex of junctions, including TJs, adherens junctions (AJs), and desmosomes [18]. TJs consist of transmembrane proteins (e.g., claudin and occludin) that seal the intercellular space and peripheral membrane proteins (e.g., zonula occludens (ZOs) and cingulin). The function of the TJs is determined by the expression level, distribution, and phosphorylation of the TJs’ proteins [68].

The claudin-2, -3, -4, -7, -12, -14, and -15 isoforms are expressed in the epithelium of the IB [69,70]. Some of these claudins (e.g., claudins 3, 4, 7, and 14) form a selective barrier to macromolecules and ions, whereas others (claudins 2, 12, and 15) are pores to ions and water. Claudin 5, on the other hand, is less expressed in epithelial cells than in endothelial cells and its role as the gatekeeper of the permeability of the IB is less important than that of the BBB [71].

Occludin, adherens junctions, and claudin molecules are linked to the cytoskeleton by zonula occludens (ZOs). The functions of ZOs are still under investigation. Most ZOs are known to regulate the assembly as well as stabilization of TJs [72]. Nevertheless, the role of ZO-1 in the TJs is still unclear. Unlike endothelial cells, recent studies found that ZO-1 is not essential for the barrier function and is not required for mucosal organization but for its repair [73]. Though the paracellular barrier properties of the IB are quite conserved throughout the whole intestine, the transcellular barrier properties of the IB differ according to the section of the intestine that is considered. The efflux of xenobiotics and their conjugates from the epithelial cells occurs through diverse membrane transporters such as the ATP-binding cassette (ABC) proteins such as P-gp (Glycoprotein P) and MRP1 and 2 (Multidrug-Resistant Protein). In the BBB, they are recognized for their ability to modulate the absorption, distribution, metabolism, secretion, and toxicity of xenobiotics. Despite the apical distribution on enterocytes, little is known regarding how ABC transporters interact with the GM. Recent studies reported that their expression is influenced by gut microbes (pathogenic or otherwise) (Figure 2) [74].

## 4. The Structure and Functions of the Blood–Brain Barrier

The BBB is a barrier localized at the level of the brain microvasculature. It is the major brain barrier in terms of the length, close to 650 km, and the surface, 10–20 m^2^ [75,76,77]. The BBB is formed by the brain endothelial cells (ECs) lining the brain microvessels.

ECs, pillars of the BBB, are in close contact with the brain pericytes sharing the same basement membrane (BM) layer and are both surrounded by a continuous sleeve of astrocytic feet. Together, with microglia and neurons, they form the neurovascular unit (NVU) and cooperate and intercommunicate to ensure the brain’s health [13,78]. Compared with other endothelial and IB epithelial cells, the morphology and functional properties of ECs are different as they lack fenestrations to limit the movement of molecules, have a low rate of vesicular transport to prevent the unspecific transport of large hydrophilic molecules to the CNS, are enriched in mitochondria, and have very few vacuoles of endocytosis [70,79,80,81]. To restrict the passage between the adjacent endothelial cells and protect the CNS from exposure to potential harmful molecules, ECs are tightly sealed through junctional complexes. The molecular composition of the BBB shows remarkable similarities to those of IB cells as they are formed by strands of occludin, JAM (junctional adhesion molecules), and claudins molecules that are linked to the cytoskeleton by ZOs (ZO-1, 2, and 3) which are membrane-associated guanylate kinases [21]. Claudin 5 is critical to the BBB formation, as claudin 5-deficient mice show a size-selective leakage of the BBB, whereas claudins 1, -3, and -12 have also been identified at the IB [70].

In contrast to IB cells, the TJs of the BBB are localized only at the apical surface of the non-fenestrated endothelium, sealing the paracellular route and making the ECs’ penetrance to the intravascular materials lower than peripheral endothelial cells [82].

The junctional complex contributes to the low paracellular permeability of the BBB by limiting the paracellular movement of endogenous and exogenous compounds. Thus, TJs are particularly important for BBB function and their loss can greatly increase its permeability, resulting in inflammation and neuropathology [83].

The BBB is an even more selective barrier than the IB. To cross the BBB, xenobiotics can diffuse passively through the membrane of ECs (transcellular route) if they are lipophilic or will be taken up by transporters. Indeed, ECs express specific transporters and receptors at the apical (blood side) and basolateral (brain side) membranes to ensure efficient nutrient supply and brain waste elimination, such as the glucose transporter (GLUT1), the monocarboxylate transporters (MCT), and the low-density lipoprotein receptor (LDL-R) that participate in the transport of glucose, ketone bodies, and lipoproteins, respectively [19,84].

In addition, ECs express several members of the ATP-binding cassette (ABC) family, such as ABCB1 (or P-gp), ABCC1 (or MRP1), and ABCG2 (or BCRP: Breast Cancer-Resistant Protein) that all impede the penetration into the brain of various potentially toxic compounds and prevent their CNS accumulation (Figure 2) [85,86]. The functional activity of these transporters increases or decreases under the influence of internal and external factors including pesticides with damage to vessels and the CNS [87,88,89].

## 5. Gut-Microbiota-BBB Communication

Many complex and bidirectional interaction mechanisms exist between the gut and brain: neural, metabolic, endocrine, and immune pathways. This complexity of interactions is summarized in the term “gut-brain axis” [90].

The neural bidirectional communication network with the GM mainly includes the enteric nervous system that controls the gastrointestinal system and the vagus nerve that allows the bidirectional signaling between the gut and the CNS [23,91]. The endocrine pathway involves a connection between the GM and the hypothalamic–pituitary–adrenal (HPA) axis through hormones. Both neural and hormonal lines of communication combine to allow the brain to influence intestinal activities and vice versa under the influence of the GM [92].

Another intercommunication exists through the bloodstream and is possible because, beyond the mucus and intestinal epithelial barrier, the gut is equipped with a gut vascular barrier (GVB), which acts as a gatekeeper to control the access of molecules and microorganisms in the systemic blood circulation [93,94]. This communication is mainly mediated by microbiota. The GM sends signals to the brain and vice versa by stimulating the release of intestinal hormones or by transforming dietary components into several substances including the amino acids, neurotransmitters (serotonin, tryptophan, and gamma-amino-butyric acid), and vitamins that influence the metabolism and the immune system, which in turn influence the integrity of the BBB and brain function [95,96].

In addition, the GM can affect the BBB through the regulating secretion of inflammatory factors and producing systemic circulating metabolites such as SCFAs by the fermentation of dietary fibers and LPS [97]. SCFAs can translocate from the intestine to the brain through the bloodstream and cross the BBB [98,99]. Some studies have reported various effects on immune cells, including Treg cells [55,100] or T effector cells and macrophages [99]. Through these metabolites and other molecules, the microbiota regulates the innate immune system and modulates the structural and functional integrity of the IB and BBB. A study showed that fetal BBB maturation is delayed in germ-free (GF) pregnant mice and the expression of TJs’ proteins claudin-5 and occludin was reduced [56]. These findings highlight the importance of the GM in a healthy barrier’s phenotype and may explain how disorders can be associated with a barrier’s physiology disturbance.

In fact, recent research has shown that GM disorders (dysbiosis and/or bacterial translocation) are often associated with a leaky IB and BBB disruption accompanied with inflammation [101,102]. Ancient medical texts referred to inflammation as a redness, warmth, pain, and functional loss, while currently it is defined as the immune system’s way of eliminating foreign substances and endogenous stress signals. The resolution of systemic inflammation results in the successful clearance of pathogens, but untreated or repeated acute inflammation could worsen the condition or transform into chronic systemic inflammation characterized by proinflammatory biomarkers [103]. In other terms, a plethora of secreted proinflammatory cytokines (IL-1b, tumor necrosis factor (TNFα), or INF-ɣ), C-reactive protein (CRP) [104], lipoteichoic acid from Gram-positive bacteria, (LPS) from Gram-negative bacteria, and double-stranded RNA from viruses will translocate, cross the IB, and reach the blood [32,105], affecting the arrangement and expression of TJs and the expression of efflux pumps and transporters of the BBB, thus impairing the barrier’s integrity. Specifically, in the case of a microbiota dysbiosis scenario, Long et al. explain that local intestinal inflammation induced by dysbiosis lies in the communication between the host and bacteria [101]. LPS-mediated communication is possible because of its recognition by the TLRs of monocytes, macrophages, and microglia at the level of both barriers, enhancing the secretion of proinflammatory cytokines. LPS activate the nuclear factor-kappa B (NF-κB) signaling pathway by binding to TLR4 [97,106]. It is this activation that is responsible for the secretion of proinflammatory cytokines which leads to systemic inflammation [101]. The resulting cocktail of proinflammatory cytokines (TNF-α and interleukin (IL)-1β and IL-6) in the bloodstream in addition to the LPS disrupt the IB and BBB (increase in the barriers’ permeability) and promote α-synuclein deposition [101,107,108]. This increase in permeability can be associated to TJs’ modulation mediated by released cytokines. TNF-α is a cytokine that has been reported to be involved in many intestinal diseases and can induce altered barrier function and increased vascular permeability. TNF-α decreased the expression of the TJs’ proteins, including ZO-1, occludin, and claudin-1, and increased the claudin-2 expression [70]. Notably, BBB dysfunction and leakage threaten brain safety, leading to brain inflammation and opening the path to the development of neurodegenerative diseases [77].

Gut dysbiosis is considered to be an important environmental factor responsible for the loss of barriers’ integrity. Any disruption of the microbiota by diet and food contaminants leading to dysbiosis, translocation, and eventually inflammation will affect the integrity of the IB and BBB. Microbiota disorders and pathways lead to inflammation and the breakdown of both barriers, particularly in the case of chronic exposure to pesticide residues.

## 6. Pesticide Residues Exposure and Effects on the Gut and BBB

As previously mentioned, one important connection between the gut and the brain is the microbiota. This means that any dysregulation of the microbiota (named dysbiosis) can affect the two parts of this axis. Oral-exposure substances that can disrupt microbiota can be in diet, drugs, antibiotics, and, most importantly in this review, pesticide residues.

Pesticide residues are food and water contaminants. Some of them alter the composition of the GM and disrupt and cross the IB [25,109,110]. They are defined by the Food and Agriculture Organization (FAO) as any substance or mixture of substances intended for preventing, destroying, or controlling any pest, including vectors of human or animal disease, unwanted species of plants or animals, causing harm during or otherwise interfering with human activities (production, processing, storage, transport, marketing of food, agricultural commodities, etc.) [111]. These substances are classified based on various criteria, such as the targeted pest organism type (fungicide, herbicide, insecticide, etc.), chemical composition (e.g., synthetic organic insecticides: organochlorines, organophosphates, carbamates, and pyrethroids), and mode of entry in the body [3,112]. The mode of action of pesticides targets the physiological systems of the pests they kill, but this can result in poisoning nontarget species, such as humans (phylogenetic similarities in digestive, respiratory, and nervous systems). Since the beginning of the 20th century, the use of pesticides and fertilizers is intensive to ensure large agriculture production to adapt with demographic growth. The most frequently used chemical families in agriculture are the synthetic organic insecticides mentioned above and triazines [5]. This means that we are not only exposed to one or two but to a mixture of environmentally persistent pesticides daily. Thus, the assessment of the impact of these substances nowadays should consider the chronic cumulative effect as well as the cocktail effect [113,114,115]. The concept of metabolization (biotransformation by microbiota bacteria and liver detoxification enzymes) is also a point of interest because the pesticide metabolite can be more harmful than the parent compound, indicating a metabolic activation. Generally, oxon-type intermediate metabolites are more hazardous than their parent pesticide. Organophosphates oxon-metabolites are an example of more toxic metabolites than the corresponding pesticide: Chlorpyrifos-oxon (CPF-oxon) is more potent than Chlorpyrifos (CPF) itself, same as for paraoxon and parathion [2,116,117]. The neurotoxic effects and impact on the gut-microbiota-BBB axis of the most detected pesticides in food (according to the EFSA European Food Safety Authority) [118] are detailed in Table 1. This review details the effects of the most used organophosphate insecticide, CPF, on the gut-microbiota-BBB axis.

### 6.1. Chlorpyrifos (CPF)

#### 6.1.1. CPF Utilizations until 2022

Organophosphorus pesticides are extensively used worldwide because of the wide range of pests they kill, the broad spectrum of applications (food trees, wheat, corn, almonds, tea, etc.), and their persistence in soil [2,136]. Chlorpyrifos (CPF) is the most commonly used thionophosphorus organophosphate insecticide, available on the market at a low price since 1965, especially in France, one of the first pesticide consumers in Europe. In 1998, only one study was sufficient for the European Union (EU) to give use approval of CPF, but it took more than hundreds of studies showing neurotoxic, metabolic, and endocrine effects to make the decision in 2019 limiting its use. Nonetheless, CPF production has not been banned. Therefore, pollution in the environment is still present. After the ban of CPF use in the EU by the EFSA, except for the culture of spinach in France, the EPA (United States Environmental Protection Agency) made a recent regulatory decision banning CPF for food uses in the United States in February 2022 (EPA final rule) [137]. However, on one hand, it is still allowed for mosquito control, on tobacco, on plantations not for feed purposes, and on food destined to export when it complies with foreign purchaser specifications (updates on CPF uses in 2022, Iowa State University) [138]. Therefore, these limitations do not prevent the accumulation of CPF in the soil and then in water and thus its uptake by aquatic species and its entry into the food chain of human beings [139]. And still, many other regions of the world including China and India continue allowing CPF on crops. On the other hand, it has been demonstrated by many studies on the biotransformation of CPF in soil that its half-life has a very wide range, from 360 days to 17 years, because its fate depends on the initial concentration used on plants and the biodegradation rate. This means that even after limitations, measures should be taken concerning the control of CPF residues in soil, local and imported food, and most importantly in human blood or urine. In addition, strict measures should be adopted in countries where residents have easy access to this dangerous pesticide: in Iran, CPF residues are found in the milk of breast-feeding mothers, their urine, and even their children’s urine [140]. A total of 92% of these mothers confirmed pesticides’ house use, an activity banned since 2001 in the US [106]. Rathod and Garg reported a scenario in India where the commonest method of suicide (40.5%) is organophosphorus compound (OPC) intake [2,141]. Therefore, limitations are essential in these countries to protect residents and their future generations.

#### 6.1.2. CPF Mechanism of Toxicity and Toxicokinetic

Chlorpyrifos is absorbed by all routes of exposure. Urinalyses of exposed human volunteers indicate that approximately 70% is absorbed by the oral route [2,142]. After CPF exposure and then distribution throughout the body, cytochrome P450 (CYP) in the liver metabolizes CPF, replacing the sulfur group with oxygen, to CPF-oxon, a metabolite that is more toxic than CPF itself. The detoxification of CPF-oxon consists of oxidase enzymes hydrolyzing it to diethylphosphate (DEP), diethyl thiophosphate (DETP), and 3,5,6-trichloro-2-pyridinol (TCP) [2]. TCP is a specific metabolite of CPF, whereas DEP and DETP can be detected after exposure to other organophosphates [143,144]. It is the metabolic bioactivation to CPF-oxon that leads to the irreversible inhibition of acetylcholinesterase (AChE) preventing the breakdown of acetylcholine (ACh), a neurotransmitter that ensures nerve cells communication. The accumulation of ACh in the synaptic cleft overstimulates the neuronal cells, leading to a collapse in the nervous system of insects (National Pesticide Information Center) [145]. Similarly, in higher vertebrae, in particular humans, CPF has a cholinergic effect and has been demonstrated to have a plethora of non-cholinergic effects.

#### 6.1.3. CPF Biotransformation by Intestinal and Soil Bacteria

The research on xenobiotics degradation in the human body mostly focuses on detoxification by CYP and other liver enzymes, but based on the literature, a minority of the studies focus on their degradation by intestinal microorganisms and specifically bacteria. A 2013 study by Harishankar and colleagues found that 70% of CPF was degraded to TCP by *Lactobacillus fermentum*, 61% to CPF-oxon by *Lactobacillus lactis*, and 16% of CPF to CPF-oxon and DEP by *Escherichia coli* [146]. Recently, insect experimental studies were conducted to investigate the fate of pesticides by an intestinal microorganism [147]. A 2021 study assessed the biodegradation of organophosphorus including CPF by isolating insect gut microbial species. Four potential bacterial endosymbionts such as *Bacillus subtilis*, *Bacillus licheniformis*, *Pseudomonas putida*, and *Pseudomonas cereus* used CPF as a unique source of carbon and energy for their growth and enzymatic function. They found that *Pseudomonas cereus* and *Pseudomonas putida* have more potential to degrade the CPF [148].

Similarly, in soils, the researchers suggested that CPF is totally degraded by microorganisms and especially by *Pseudomonas putida* MB285 to form the primary products TCP and DETP, which are further decomposed into non-toxic metabolites, such as CO_2_, H_2_O, and NH_3_ [149]. *Pseudomonas* is a diverse genus with multiple degradation pathways. It was reported that *Pseudomonas putida* MAS-1 had the highest degradation efficiency for CPF in *Pseudomonas genus*, with a 90% degradation rate within 24 h [150]. Furthermore, another study published in 2008 reported that different bacteria contribute to the biodegradation of CPF in five aerobic consortia, based on antibiotic resistance survival and REP-PCR (Repetitive Extragenic Palindromic Polymerase Chain Reaction). The results illustrated that 75–87% of the CPF was degraded to TCP after 20 days of incubation by *Pseudomonas aeruginosa*, *Pseudomonas fluorescence*, *Bacillus subtilis*, *Brucella melitensis*, *Klebsiella* sp., *Bacillus cereus*, and *Serratia marcescens*. However, the results also showed that the TCP disappeared after 30 days of incubation [151]. A mini review published by Supreeth in 2017 evaluated the biotransformation of CPF and endosulfan by bacteria and fungi and discussed the aftereffects of their transformed byproducts (metabolites) [136]. The degradation of CPF to TCP was executed by different bacteria species, *Enterobacter* sp., *Stenotrophomonas* sp., *Sphingomonas* sp. [152], *Pseudomonas aeruginosa*, *Bacillus cereus*, *Klebsiella* sp., *Serratia marscecens* [153], *Bacillus subtilis inaquosorum* strain KCTC13429, *B. cereus* ATCC 14579, and *B. safensis* F0-36b [154], and to CPF-oxon 3,6-Dihydroxypyridine-2,5-dione and DETP by *Pseudomonas putida* (NII 1117) *Klebsiella* sp. (NII 1118) *P. stutzeri* (NII 1119), *P.aeruginosa* (NII 1120) [155], and *Ralstonia* sp. Strain T6 [156].

#### 6.1.4. CPF Effects on Gut-Microbiota-BBB Axis

As the gut and microbiota have an impact on the absorption and metabolization of pesticides, these pesticides can have serious effects on all parts of the gut–brain axis. Prenatal pesticide exposure studies detected parent molecules of organophosphates and/or their metabolites, CPF in particular, in meconium [157,158,159,160]. In fact, meconium has become a biomarker of in utero pollutants exposure. At this stage of rapid growth and development, especially of the brain, humans are more vulnerable and sensitive to the toxic effects of pesticides [161,162]. The message behind these analyses is that even before birth, which means before gut microbial colonization, we are exposed to CPF, which seems to cross the placental barrier. In addition, considering the oral route, before reaching the brain to inhibit acetylcholinesterase, the first organ to encounter CPF or any food contaminant is the digestive tract. Thus, it is necessary to investigate the organophosphate effects on the intestinal tract. Working on the rat offspring NOAEL of 1 mg/kg/day, Joly et al. studied the effect of 1 mg/kg/day or 5 mg/kg/day of CPF perinatal exposure on the GM of the rats’ progeny at two developmental time points: weaning (D21) and adulthood (D60) [34]. Their results show that CPF exposure induces microbial dysbiosis: a decrease in *Lactobacillus* spp. counts in the ileum, cecum, and colon as well as a decrease in *Bifidobacterium* spp. at D21 in the ileum and D60 in the colon, and an increase in *Clostridium* spp. and *Staphylococcus* spp. counts in the cecum and colon at D21. This means that CPF exposure reduces potentially beneficial bacteria and increases potentially pathogenic ones. The epithelial thickness of the ileum and colon was decreased by CPF treatment. Because the first study was based on classical microbiology tests, they researched further to identify more specifically intestinal bacteria by MALDI-TOF-MS and investigate bacterial translocation from intestinal segments to sterile organs [30]. By molecular typing, they were able to confirm the translocation of *Staphylococcus aureus* to adipose tissues, kidney, and Peyer’s patch from 12.5% of CPF-exposed rats’ intestinal segments and a 5% translocation of *Enterococcus faecalis* to the liver. This is logically explained by the increase in gut permeability induced by the decrease in ZO-1 and claudin 4 transcriptional expression in the ileum and colon, especially on D21 [62]. CPF exposure of an in vitro artificial human intestine (SHIME^®^) combined with a Caco-2/TC-7 model was associated with a decrease in the tight junction gene expression, occludin and ZO-1, and an increase in the proinflammatory chemokine interleukin-8 (IL-8) [121]. Zhao et al. also confirmed abnormal intestinal permeability in CPF-exposed mice and reported microbiota dysbiosis (a decrease in *Lactobacillaceae* and *Firmicutes* and an increase in *Bacteroidaceae* and *Bacteroides*) and alterations in the metabolism of SCFAs that led to intestinal inflammation [109]. This is an expected result because CPF treatment alters the microbial community that produces these metabolites. In addition, an analysis of serum showed an increase in LPS by CPF treatment. J. W. Li and colleagues showed that chronic exposure to 0.3 mg CPF/ kg body weight/day induced a significant increase in TNF-α and IL-6 in the serum of exposed rats [163]. These observations might explain the alterations in the functional integrity and structure of the BBB by CPF, highlighted by Parran et al. who worked on an in vitro BBB model (bovine endothelial cells and neonatal rat astrocytes) [87]. Another study on an in vitro BBB model (rat brain endothelial cells and neonatal rat astrocytes) showed that the short-term CPF treatment at low concentrations alters the expression levels of the claudin 5, ZO-1, and TRPC4 (transient receptor potential canonical channels) genes, disrupting the BBB integrity (TRPC regulates the calcium influx that modulates the paracellular permeability of the endothelial cells of the BBB) [88]. The claudin 5 gene is the main tight junction gene involved in the BBB tightness and ZO-1 ensures the support to the TJs’ architecture [164,165]. Thus, CPF targets the most important actors of the BBB integrity.

Because the exposure to pesticides is often accompanied with saturated fats and refined sugars (an unbalanced diet based on fast food and processed food), other studies assessed the impact of the association of CPF to a High Fat Diet (HFD) on intestinal microbiota and the IB. Guibourdenche and colleagues confirmed that the chronic perinatal exposure to the NOAEL dose of CPF alone induced a decrease in transcriptional TJs expression and also demonstrated an increase in proinflammatory cytokines and is aggravated by the association to an HFD [31,122].

These experimentations all together point to the impact of CPF on the entire gut-BBB axis even though its main mechanism is the inhibition of acetylcholinesterase in neurons. Further investigations are necessary to identify molecular pathways that explain the barriers’ disruption by organophosphates.

#### 6.1.5. CPF Molecular Pathways Underlying Its Effects

In parallel to studies analyzing the effects of organophosphates on the gut–brain axis, there are several studies investigating the molecular pathways associated to them. It has been demonstrated that organophosphates induce apoptosis by affecting signaling molecules, including c-Jun NH2-terminal protein kinase (JNK), p38 MAP kinase, and extracellular signal regulated protein kinase (ERK1/2) [166,167]. ERK1/2 is activated by neurotrophic factors and growth factors, whereas environmental stresses such as reactive oxygen species (ROS) activate JNK and p38 MAP kinases. The activation of JNK and p38 MAP kinases induces apoptosis, while the activation of ERK1/2 is protective against it [168]. In fact, CPF increases JNK, ERK1/2, and p38 MAPK phosphorylation [169]. It is known that mitogen-activated protein kinases (MAPKs) regulate matrix metalloproteinases (MMPs). MMP9 is regulated by ERK1/2 [170]. MMP9 activation yet increased the IB permeability in a Caco-2 in vitro model [171]. MMP9 upregulation leads to BBB leakage as well, through the rearrangement and/or degradation of the tight junctions [172,173]. Moreover, MAPKs have a role in the inflammatory response. Interestingly, CPF can upregulate cyclooxygenase 2 (COX-2) through MAPK activation [169]. In addition, CPF upregulates inflammation-related genes and the protein level of NF-κB and TNF-α [174] which in turn can deteriorate the BBB permeability [84]. A possible explanation for the CPF implication in these pathways is as follows: CPF increases NADPH oxidases (NOXs) and superoxide levels which increase ROS signaling and oxidative stress in cells [175]. The increase in ROS signaling induces, on one hand, the upregulation of TNF-α and NF-κB and, on the other hand, the increase in the ASK1 expression responsible for the activation of JNK upregulation which is one of the MAPKs that regulates MMPs.

## 7. Beneficial Modulation of the Gut-Microbiota–Brain Axis

Recent studies on pesticide effects are focusing on a new concept: the recovery. In fact, a study of the effect of different doses of imazalil on mice GM evaluated the time of recovery after 2 and 15 weeks of exposure. They found that 30 or 45 days after impregnation with this fungicide, the bacterial composition at the phylum level recovered to the control level [134]. This highlights the fact that a change in the microbiota composition is flexible and tends to regain balance. In addition, new clinical trials indicated that the gut-microbiota–brain axis pathways and mechanisms are prone to dietary modulation and are of vital interest in clinical nutrition. As a matter of fact, dietary interventions and supplementation with probiotics and prebiotics can reshape the bacterial composition and are now administered as “psychobiotics” to treat neurological disorders because of their beneficial effects on the brain [176]. This means that by reshaping the GM, the whole gut-microbiota-BBB axis is positively modulated. The concept of probiotic use in the modulation of gut microflora was initiated in the 19th century by Elie Metchnikoff who theorized that “health and longevity could be achieved by manipulating intestinal microflora, i.e., replacing harmful microbes with beneficial microbes” while prebiotics were introduced by Gibson and Roberfroid in 1995 [177].

Their supplementation is now considered a promising approach that alleviates the negative effects of food contaminants [178]. In effect, bacterial strains that are considered probiotic strains (mostly *Lactobacillus* strains) can bind to xenobiotics and reduce their toxicity (through biotransformation) and the amount absorbed by the host [179]. *Lactobacilli* spp. are known to have the highest anti-inflammatory effects [52]. To exert their beneficial roles, probiotic strains need substrates. Prebiotics partially provide probiotic strains with substrates that are “nondigestible food ingredients which selectively stimulate the growth and activity of beneficial bacterial species already implanted in the colon, and thus improve the health of the host” [180]. Prebiotics are fructo-oligosaccharides (FOS) such as inulin, galacto-oligosaccharides (GOS), trans-galacto-oligosaccharides (TOS), and resistant starch which can be found in many fruits, vegetables, grains, and milk [181,182].

Arabinoxylo-oligosaccharides and inulin induced an increase in some SCFAs (acetate, propionate, and butyrate) and a shift in the microbial composition from *Firmicutes* to *Bacteroidetes* [183]. Similarly, Sialyllactose (isolated from milk) and GOS induced the differentiation of the epithelial cells in the Caco-2 model, the modulation of the microbial composition (an increase in *Bacteroides* and *Bifidobacteria*), and consequently the production of SCFAs [184]. In fact, the fermentation of prebiotics by gut bacteria produces SCFAs which can reach the bloodstream by diffusing through gut enterocytes and have beneficial effects on the host [181]. In effect, an in vitro study linking the Caco-2 model and SHIME demonstrated that arabinogalactan and FOS decreased proinflammatory cytokines (IL-6 and IL-8), increased the anti-inflammatory cytokine IL-10, and improved the gut barrier permeability (the TEER measurements of the Caco-2 model) [185]. These findings emphasize the direct effect of prebiotics on IB function through the modulation of the microbiota composition. However, these oligosaccharides and the products of their fermentation (SCFAs) can improve IB function through TJs modulation [186]. Many studies underline the role of inulin [187] and FOS [188] in enhancing IB through TJs assembly.

Because they have a beneficial effect on the IB and microbial composition, researchers investigated if pretreatment or supplementation with prebiotics could attenuate damage in an inflammatory environment or intestinal harm caused by food contaminants exposure. As an example, GOS pretreatment can alleviate damage of the IB in an inflammatory environment (in LPS-challenged mice) [189]. Additionally, inulin supplementation to rats exposed to CPF, an HFD, or the association of both CPF and a HFD and in in vitro models (SHIME and Caco-2) reversed their effects on microbial composition by increasing potentially beneficial flora (*Lactobacillus* and *Bifidobacterium*), decreasing potentially pathogenic ones (*Enterococcus* and *Enterobacteriacea*) and improving the IB integrity [26,120,121,122]. Thus, prebiotics use through nutrition can modulate the GM with beneficial outcomes. New methods are now emerging such as fecal transplantation, a strategy to deal with dysbiosis, that has been shown to have a positive effect on the treatment of Parkinson’s disease [190].

## 8. Conclusions

Pesticides are generally considered to contribute to global food security, although the extent of this contribution and how it is balanced against their potential to harm human health is the subject of intense public debate. There is now evidence that the use of certain pesticides, such as Chlorpyrifos and its metabolites, has serious and long-term negative effects not only on the environment but also on human health. However, the effects of pesticides on the functional barriers that protect our body are not well known, either for the general population or for so-called vulnerable populations, such as pregnant women and their offspring. The intestinal barrier, the blood–brain barrier, and the gut microbiota play an important role in the absorption and access of food contaminants. The literature demonstrates that pesticide residues and saturated fats disrupt the intestinal microbial balance and the IB and BBB structure and function. Because the GM is the most important part ensuring the communication between the intestinal tract and the brain within the gut-microbiota-BBB axis, it is becoming one of the targets to palliate food contaminants effects and treat intestinal and brain diseases. In fact, certain prebiotics could be beneficial and counteract their effects by adjusting the microbial balance and improving the functions of biological barriers. Nevertheless, little is known about these preventive effects at the level of the gut–brain microbiota axis and still many questions are unanswered. Thus, further studies on the modulation of the GM are necessary to establish and evaluate preventing nutrition methods based on prebiotic use and other types of methods, such as fecal transplantation.

Finally, considering the DOHaD concept stating that individuals are more vulnerable to food contaminants during the perinatal period, raising awareness in the population, especially the most vulnerable categories, should be implemented by promoting a diet rich in natural fibers and lacking food contaminants.

## Figures and Tables

**Figure 1 ijms-24-06147-f001:**
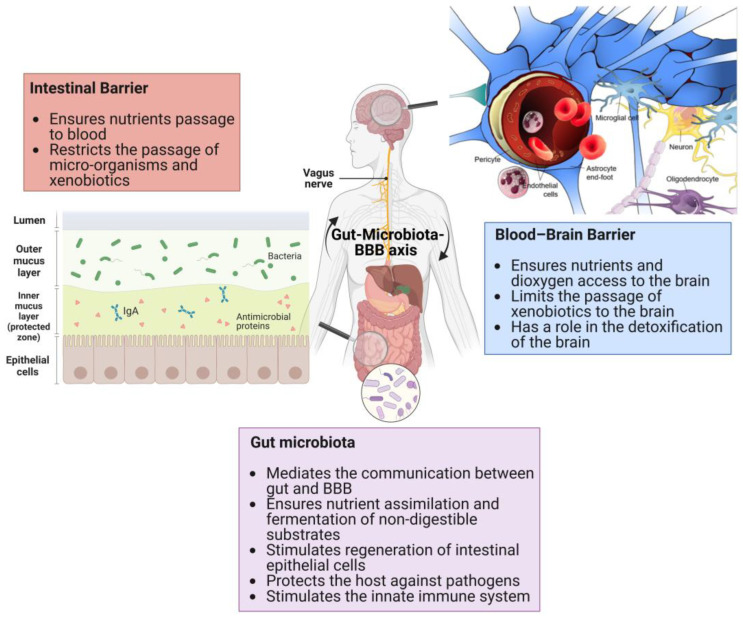
Gut-Microbiota–Blood–Brain Barrier Axis.

**Figure 2 ijms-24-06147-f002:**
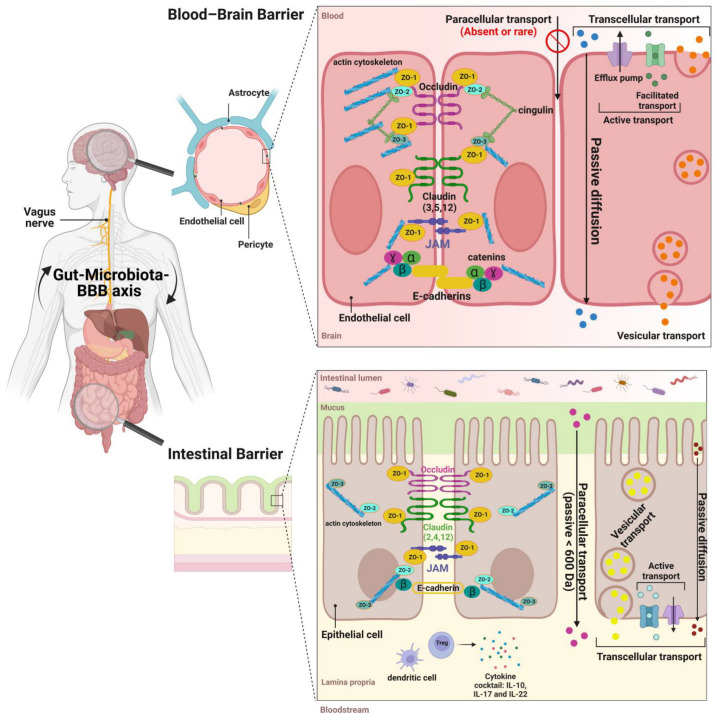
Similarities and differences between the intestinal barrier (IB) and blood–brain barrier (BBB). The IB is made up of epithelial cells separating the gut lumen from the internal space, whereas the BBB is formed by endothelial cells (ECs) lining the blood vessels separating lumen of blood vessels from the central nervous system (CNS) parenchyma. The first defense component of the IB, separating the microbiota from the epithelial cells, is the intestinal mucosa: mucus layer composed of highly glycosylated mucin proteins, defensins, Immunoglobulin A (IgA), and the inner lamina propria where T-regs (regulatory T cells) and resident dendritic cells maintain an anti-inflammatory environment by secreting an appropriate cocktail of cytokines. Each barrier tightly regulates the movement of molecules and ions between the cellular spaces through paracellular (rare in the BBB) and transcellular transports (diffusion, vesicular, and active). The active transport is ensured by efflux pumps and specific transporters. Thus, these two barriers are considered physical, immunological, and dynamic barriers protecting the host against food contaminants and pathogens. The molecular composition of BBB endothelial tight junctions, ensuring paracellular passage restrictions, shows remarkable similarities to those in epithelial cells (but more stringency at the BBB) as they are formed by strands of occludin, claudin molecules, and JAM (junction adhesion molecules) that are linked to the cytoskeleton by zonula occludens (ZO-1, 2, and 3) as well as intracellular proteins (e.g., catenin).

**Table 1 ijms-24-06147-t001:** Neurotoxic effects and impact of pesticides on the gut-microbiota-BBB axis. AChE: acetylcholinesterase; BBB: blood–brain barrier; C57BL/6: C57 black cellCaE: carboxylesterase; ChE: cholinesterase; CPF: Chlorpyrifos; IB: intestinal barrier; IL: interleukin; IMZ: imazalil; IFN-ɣ: Interferon gamma; ppm: parts per million; Lcn-2: Lipocaline 2; N2a: mouse neuroblastoma cells; PC12: rat Pheochromocytoma cells; SHIME: Simulator of Human Intestinal Microbial Ecosystem; TEER: transendothelial electrical resistance; TJs: tight junctions; TNF-α: tumor necrosis factor; TRPC4: transient receptor potential canonical channels; ZOs: zonula occludens. ↓ and ↑ symbols refer to decrease and increase, respectively.

Pesticides	Effect on Gut	Effect on Microbiota	Effect on BBB	Neurotoxic Effect	References
**Organophosaphates**					
Chlorpyrifos (CPF)	**a.** ↓ in epithelial thickness of ileum and colon of rats after exposure to 1 mg/kg/day of CPF ↓ of tight junction gene expression of the intestinal barrier and ↑ of proinflammatory cytokines	**c.** Exposure to 1 mg/kg/day of CPF-induced (1) microbial dysbiosis in pregnant rats and offspring, a ↓ in potentially beneficial flora and an ↑ of the potentially pathogen one, and (2) bacterial translocation to sterile organs	**d.** Altered gene expression levels of claudin 5, ZO-1, and TRPC4 genes disrupting the barrier integrity in an in vitro BBB model	**e.** Exposure to 1.5 (low dose) and 3 (high dose) mg/kg/day of CPF-induced inhibition of cholinesterase in rats in a dose-related manner **f.** Inhibition of carboxylesterase (CaE) and cholinesterase (ChE) activities by 43–100% in an vitro BBB model treated with (0.1 to 10 µM) of CPF	**a.** [31,34,35]**b.** [119,120,121]**c.** [26,30,34,122] **d** and **f.** [87]**e.** [123]
	**b.** ↓ of TJs gene expression of the IB (Caco-2/TC-7 model treated with SHIME supernatant exposed to 3.5 mg of CPF for 30 days)			**g.** Exposure of pregnant rats to 1 and 5 mg/kg/day of CPF-induced inhibition of AChE of juvenile and adult offspring leading to high sleep apnea index	**g.** [124]
Diazinon		**h.** Exposure to 4 ppm for 13 weeks in drinking water had an impact on bacterial populations and composition of *Lachnospiraceae*, *Ruminococcaceae*, *Clostridiaceae*, and *Erysipelotrichaceae* in male mice		**i.** Exposure to 0.5 and 2 mg/kg/day induced long-lasting alterations in cognitive function in adolescence and extending into adulthood of rats	**h.** [125,126,127]**i.** [128]
Malathion		**j.** ↓ in bacterial populations (depletion of 4 genera) in male mice exposed to 2 mg/mL in drinking water for 13 weeks	**k.** ↓ in the TEER in two BBB in vitro models treated with malathion (10^−3^ to 10^−8^ M)	**l.** Induces neurotoxicity through ChE inhibition and non-cholinergic mode: apoptotic cell death by targeting mitochondria in N2a neuroblastoma mouse cells (at 0.25, 0.5, or 1 mM for 8 h)	**j.** [125,127] **k.** [129] **l.** [130]
**Herbicides**					
Glyphosate	**m.** ↑ in proinflammatory cytokines transcriptomic expression (IL-1β, IL-6, and TNF-α) after exposure to 5, 50, and 500 mg/kg for 35 days in male rats	**n.** ↓ in the relative abundance of *Firmicutes* and *Lactobacillus* but increased *Fusobacteria* in male rats exposed to 500 mg/kg for 35 days	**o.** ↑ in barrier permeability to fluorescein (at 1 and 10 µM of glyphosate) and ↓ in claudin-5 fluorescence intensity (at 100 and 1000 µM) after 24 h treatment of the BBB model	**p.** 24 h of treatment with high doses of glyphosate (100 µM) can affect neurons metabolic activity	**m** and **n.** Q. [110]**o** and **p.** [131]
**Fungicides**					
Imazalil	**q.** ↑ in a colonic inflammation biomarker (Lcn-2)↑ in mRNA levels of TNF-α, IL-1β, IL-22, and IFN-ɣ in the colon after exposure to 100 mg/kg bw/day IMZ for 28 days in mice	**r.** ↓ in *Bacteroidetes*, *Firmicutes*, and *Actinobacteria* in the colon after exposure to 100 mg/kg bw/day IMZ for 28 days in mice		**s.** ↑ in oxidative stress ↑basal calcium ions Ca^2+^ (indicating inhibition of depolarization-evoked calcium influx) in an in vitro model of rat dopaminergic PC12 cells treated with 100 µM of IMZ	**q** and **r.** [25]**s.** [132,133]
	**t.** 15 weeks administration of IMZ at doses of 0.1, 0.5, and 2.5 mg/kg/day to C57BL/6 mice led to:		**u.** 48% inhibition of AChE by IMZ (at 500 µM, in vitro enzymatic inhibition assays)	**t.** [134] **u.** [135]
	IB dysfunction	gut-microbiota dysbiosis

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
