# Peer review of "Impact of Pesticide Residues on the Gut-Microbiota–Blood–Brain Barrier Axis: A Narrative Review"

_ijms, 2023, doi:10.3390/ijms24076147_

Round 1
Reviewer 1 Report
The authors presented an original and well-written manuscript entitled “Gut-Microbiota-Blood-brain barrier axis: A narrative review” with the aim to address the issue of chronic exposure to low-level of pesticides and its effect on the human gut-microbiota-blood-brain barrier (BBB) axis. First, the authors discuss the similarities and differences between the IB and the BBB. Then they focus on current knowledge of the effects of pesticides on this axis and raise awareness of the danger of chronic exposure, especially during the perinatal period. Finally, the authors briefly discuss whether prebiotics could counteract the effects of these xenobiotics. The main focus of this review is an organophosphate pesticide chlorpyriphos.
This topic is crucial since the use of pesticides is inevitable nowadays. Although some pesticides are banned in many countries, their production is still allowed, contributing to environmental pollution. The manuscript is very well prepared. It covers all critical aspects of mentioned issue. The literature is abundant and up to date. I have some minor comments/suggestions:
- Figures 1 and 2 are not readable in some parts. They should be enlarged.
- Lines 237 and 238: there are some unnecessary dashes. Please check.
- Line 339: Please, provide the classification of pesticides according to the mode of entry also.
- Page 11, Section 6.1. The authors should also acknowledge that in the EU, the use of chlorpyrifos is banned, but the production is not. Therefore, pollution in the environment is still present.
- The conclusion should be given in more detail.
The manuscript should be accepted after addressing the minor issues listed above.
Reviewer 2 Report
Thank you for allowing me to review this engaging and well-written review on a topic of great interest. The study evaluated the effect that xenobiotics exposure, particularly pesticides, has on the gut microbiota and the gut microbiota - BBB axis. The article combines the most remarkable findings of current work and presents them in an orderly and very intuitive manner. However, some improvements that could be made to the article are as follows:
1. A critical term when discussing gut microbiota is bacterial translocation. However, it needs to be described in the introduction. It would be interesting to introduce a line indicating what bacterial translocation consists of.
2. The authors state between lines 56-59, "The gut microbiota (GM) is a real organ system." It is necessary to clarify what they mean by this. I would appreciate this clarification on whether they speak literally or figuratively and what they mean by this.
2. In lines 96-97, the authors indicate that the colonization process is influenced by the vaginal microbiota of the mother and give as reference 34. I believe this article refers more to the type of birth than the vaginal microbiota. Please review this article and consider rewriting or deleting this sentence.
3. I recommend that the authors replace line 155 Tregs with regulatory T cells (Tregs) to better understand this part of the text.
4. For consistency of the text, I ask the authors to replace GI tract pot GIT in line 158.
5. Between lines 348-352, the authors state: "The concept of metabolization (biotransformation by microbiota bacteria and liver detoxification enzymes) is also a point of interest since the pesticide metabolite can be more harmful than the parent compound, indicating a metabolic activation (studies showed that Chlorpyrifos-oxon (CPF-oxon) is more potent than Chlorpyrifos (CPF) itself) [2,110]. "However, this statement can be extended to other pesticides such as parathion. Generally, oxon-type intermediate metabolites are more hazardous than their parent pesticide. Therefore, I request rewriting these lines to clarify this aspect.
6. I would ask the authors to recheck the literature to introduce in section 7 some mention of fecal transplantation as a strategy to deal with dysbiosis. For example, it has been shown that this technique can have a positive effect on the treatment of Parkinson’s disease, as described in the following study published in 2019:
Smith, L. M., & Parr-Brownlie, L. C. (2019). A neuroscience perspective of the gut theory of Parkinson's disease. The European journal of neuroscience, 49(6), 817-823. https://doi.org/10.1111/ejn.13869
Once all these aspects have been reviewed, this review will be ready to be published in this journal.
Reviewer 3 Report
In this review, Diwan and colleagues discuss evidence concerning pesticides and gut-microbiota-blood-brain barrier axis. The manuscript is well-written and well-organized. However, some minor points that could improve this manuscript.
- The title is too general. Because the Authors focused their attention on the effects of pesticides, they should change the title that must include pesticides.
- I have found interesting the concept of altered BBB permeability induced by dysbiosis. In this respect the Authors could include new studies that have discovered new gut derived metabolites able to affect BBB integrity. Indeed, it has been recently showed that gut microbiota alterations boosted the production of toxic metabolites such as p-cresol (PMID: 36400332), which is able to disrupt BBB integrity (PMID: 35596559; doi: https://doi.org/10.1101/2022.11.12.516113). Moreover, another study reports that the gut-derived metabolite 4-ethylphenyl sulfate (4EPS) alters brain activity probably disrupting BBB permeability (PMID: 35165440). These findings could be inserted at lines 144 or 300-305.
